# Bias in nutrition-health associations is not eliminated by excluding extreme reporters in empirical or simulation studies

Nao Yamamoto[1†], Keisuke Ejima[2,3*†], Roger S Zoh[2], Andrew W Brown[4,5]

[1]School of Human Evolution and Social Change, Arizona State University, Tempe, United States; [2]Department of Epidemiology and Biostatistics, Indiana University School of Public Health-Bloomington, Bloomington, United States; [3]Lee Kong Chian School of Medicine, Nanyang Technological University, Singapore, Singapore; [4]Department of Biostatistics, University of Arkansas for Medical Sciences, Little Rock, United States; [5]Arkansas Children's Research Institute, Little Rock, United States

**Abstract** Self-reported nutrition intake (NI) data are prone to reporting bias that may induce bias in estimands in nutrition studies; however, they are used anyway due to high feasibility. We examined whether applying Goldberg cutoffs to remove 'implausible' self-reported NI could reliably reduce bias compared to biomarkers for energy, sodium, potassium, and protein. Using the Interactive Diet and Activity Tracking in the American Association of Retired Persons (IDATA) data, significant bias in mean NI was removed with Goldberg cutoffs (120 among 303 participants excluded). Associations between NI and health outcomes (weight, waist circumference, heart rate, systolic/diastolic blood pressure, and VO2 max) were estimated, but sample size was insufficient to evaluate bias reductions. We therefore simulated data based on IDATA. Significant bias in simulated associations using self-reported NI was reduced but not completely eliminated by Goldberg cutoffs in 14 of 24 nutrition-outcome pairs; bias was not reduced for the remaining 10 cases. Also, 95% coverage probabilities were improved by applying Goldberg cutoffs in most cases but underperformed compared with biomarker data. Although Goldberg cutoffs may achieve bias elimination in estimating mean NI, bias in estimates of associations between NI and outcomes will not necessarily be reduced or eliminated after application of Goldberg cutoffs. Whether one uses Goldberg cutoffs should therefore be decided based on research purposes and not general rules.

## Editor's evaluation

This important study estimated the bias in nutrition-health associations and the reduction of bias using Goldberg cut-offs. The evidence supporting the study claims is solid, and the findings will be of interest to epidemiologists and nutrition scientists who are concerned with the effects of measurement error in health-diet research.

## Introduction

Measuring nutrition intake plays a pivotal role in obesity and nutrition studies. However, objectively measuring nutrition intake is not easy. Under natural settings with less restriction, two approaches are often used: self-report and biomarkers. Biomarker approaches are considered more accurate compared with self-report; however, they are typically expensive and require special equipment for

**\*For correspondence:** keisuke.ejima@ntu.edu.sg

[†]These authors contributed equally to this work

each nutrient. Self-report approaches are less expensive and do not require special laboratory equipment or techniques, and can be applied to the measurement of any nutrition components from macronutrients (i.e. carbohydrates, fat, and protein) to micronutrients (i.e. minerals and vitamins); however, self-reported data are known to be fraught with bias (*Schoeller et al., 1990*; *Dhurandhar et al., 2015*).

Consider the example of biomarker versus self-report energy intake. Energy intake (EI) is defined as a sum of the total metabolizable energy content of consumed foods and beverages (*Astrup, 2005*). EI can be measured using biomarkers, especially by energy balance approaches in which respiratory gas analysis or doubly labeled water (DLW) are used for the calculation of energy expenditure (EE) (*Ferrannini, 1988*). EI can be calculated using EE accounting for change in energy stores (i.e. body composition) during the period of measurement. The respiratory gas analysis typically requires the participants to wear a gas mask or stay in a metabolic chamber (*Leonard, 2012*), which restricts participants' behavior; thus, the measured EI using the respiratory gas analysis may not reflect EI from habitual eating behavior. DLW requires the participants to visit the laboratory to drink labeled water enriched with stable hydrogen and oxygen isotopes and collect urine for measurement of isotope levels. It is considered to reflect habitual eating behavior and widely accepted as a standard and reliable approach in measuring EI in free-living individuals (*Schoeller et al., 1986*; *Speakman, 1997*; *Schoeller and van Santen, 1982*). However, given the high cost of isotopes and laboratories to analyze collected urine, DLW is not practical in large-scale epidemiological surveillance (*Banna et al., 2017*).

EI measured using self-report approaches (EI$_{SR}$) is computed by summing up the energy intake from foods and beverages reportedly consumed during the period of measurement, which can commonly range from a day to a year depending on approaches (e.g. 24 hr recall, food diary, and food frequency questionnaire; *Leung et al., 2017*; *Sugiyama et al., 2014*; *Oftedal et al., 2017*). Given that self-report approaches are low cost and scale-up relatively easily, EI$_{SR}$ is still widely used in epidemiological studies. However, EI$_{SR}$ is biased compared with the EI measured by biomarker such as DLW (EI$_{BIO}$) (*Schoeller et al., 1990*) and is discouraged from use to estimate actual EI (*Dhurandhar et al., 2015*). The mechanism behind the reporting bias in EI$_{SR}$ has been argued elsewhere. For example, EI$_{SR}$ is computed from the information of reported foods and beverages using tables of food and beverage nutrients, composed of weights, calories, and the amount of macro- and micronutrients. Compositions of foods and beverages are different among brands, stores, and seasons (*Martelli et al., 2018*), and thus nutrient databases cannot be exhaustive. Consequently, EI$_{SR}$ using such databases may not be accurate (*McHenry et al., 1945*). Further factors, including social desirability (i.e. people do not want to be seen as a 'big eater' in general) (*Heymsfield et al., 1995*) and weight/social status (*Heymsfield et al., 1995*; *Johansson et al., 1998*; *Macdiarmid and Blundell, 1998*; *Johansson et al., 2001*; *Klesges et al., 1995*), are known to be associated with reporting bias. Thus, the difference between EI$_{BIO}$ and EI$_{SR}$ is due to systemic biases rather than a random error.

Despite concerns about the accuracy of self-report approaches, they have been predominantly used in large-scale nutrition epidemiology due to their high feasibility. Approaches to mitigate the bias in EI$_{SR}$ have been argued, including commonly used approaches to exclude the data of EI$_{SR}$ that are considered unreliable. For example, one approach is to exclude those reporting EI$_{SR}$ out of the range considered by the researchers to be plausible, such as from 500 to 3500 kcal/day (*Banna et al., 2017*). However, this rule does not consider individual variability and is not appropriate for those whose intake is truly above 3500 kcal/day, such as elite athletes or those whose body size is large and requires more than 3500 kcal/day to maintain their weight (*Santos et al., 2014*; *DeLany et al., 2013*).

Other approaches are to use EE prediction equations to account for individual variability in EI (*McCrory et al., 2002*; *Huang et al., 2005*; *Vinken et al., 1999*; *Livingstone and Robson, 2000*; *Xie et al., 2003*; *Briefel et al., 1995*; *Huang et al., 2004*). If the EE predicted considering various factors (e.g. age, weight, sex) is far from EI$_{SR}$, those individuals are considered under/overreporting their actual EI. Indeed, such an approach has evidently reduced misclassification (*Tooze et al., 2012*). Among proposed approaches, the Goldberg cutoffs *Goldberg et al., 1991* have been used in nutrition epidemiology to assess nutritional status in specific populations and investigate the association between nutritional status and health and socioeconomic outcomes (*Livingstone and Robson, 2000*; *Xie et al., 2003*; *Briefel et al., 1995*; *Huang et al., 2004*; *Martin et al., 2018*). Note that nutritional status in this context means dietary intakes in general; not only EI but also nutritional constituents (vitamin, fish oil, etc.) and dietary styles (Mediterranean, vegan, etc.; *Grimes et al., 2013*;

*Molina-Montes et al., 2017*; *New et al., 2000*; *Shaheen et al., 2001*; *O'Sullivan et al., 2011*). There-fore, the rationale of using the Goldberg cutoffs (and other similar approaches) to exclude data from individuals considered to have unreliable EI$_{SR}$ is based on an assumption, 'if total EI is underestimated, it is probable that the intakes of other nutrients are also underestimated' (*Livingstone and Black, 2003*). The corollary for overestimation may also be true.

Bias-correcting approaches like the Goldberg cutoffs are frequently used in nutrition studies. By bias, we are referring to the difference between the estimates and their true values. Here, we focus on two typical estimands in nutrition epidemiology: the mean of nutrition intake, and the associations between nutrition intake and health outcomes in a population. While reducing bias in the mean nutri-tion intake is useful for determining the true value of population nutrition intake from self-reported data, it is also important to reduce bias in the associations between nutrition intake and health outcomes. However, whether the Goldberg cutoffs reduce bias has not been proved theoretically or empirically. In our previous study, we demonstrated that the Goldberg cutoffs do not necessarily elim-inate the bias in estimating the associations between EI and various health outcomes (*Ejima et al., 2019*). Extending our previous study, we consider nutritional intake beyond EI to include biomarkers of sodium, potassium, and protein with the primary goal to examine if the Goldberg cutoffs eliminate or reduce the bias using a single dataset from the Interactive Diet and Activity Tracking in the Amer-ican Association of Retired Persons (IDATA) study.

**Table 1.** Variables, data generation models, and metrics of performance of the Goldberg cutoffs.

| Variables and parameters | Description |
|---|---|
| $NI_{SR}$ | Self-reported nutrition intake, where NI can be energy intake (EI), sodium intake (SI), potassium intake (PoI), or protein intake (PrI) |
| $NI_{BIO}$ | Biomarker nutrition intake |
| $NI_G$ | Self-reported nutrition intake accepted by the Goldberg cutoffs |
| $HO$ | Health outcomes, where HO can be body weight (BW), waist circumference (WC), heart rate after fitness test (HR), resting systolic blood pressure (SBP), resting diastolic blood pressure (DBP), and VO2 max (VO2) |
| $\beta_{NI,HO}$ | Estimated coefficients in the regression model using NI and HO as a dependent and an independent variables. |
| $\beta_1$ | True coefficients, the point estimate from the IDATA (used in simulation) |
| **Data generation models** | **Description** |
| $HO = a_0 + a_1 NI_{BIO} + \epsilon$ | $\epsilon$ is the error of health outcomes $\sim N\left(0, \eta^2\right)$ |
| $NI_{SR} = NI_{BIO} + e$ | $e$ is the reporting error $\sim N\left(\mu, \sigma^2\right)$ |
| $\mu\left(p\left(NI_{BIO}\right)\right) = \sum_{k=0}^{K} m_k p\left(NI_{BIO}\right)^k$ | $\mu$ is the mean error function, determined by a polynomial function of percentiles of NI$_{BIO}$ |
| **Metrics of performance of the Goldberg cutoffs (data analysis)** | **Description** |
| $b = \dfrac{\beta_{NI_{SR},HO} - \beta_{NI_{BIO},HO}}{\beta_{NI_{BIO},HO}} * 100$ | Percent bias (%) in the estimation of NI-HO association when using $\beta_{NI_{SR},HO}$. |
| $r = \dfrac{\beta_{NI_G,HO} - \beta_{NI_{BIO},HO}}{\beta_{NI_{BIO},HO}}$ | Percent remaining bias (%) in the estimation of NI-HO association when using $\beta_{NI_G,HO}$ instead of $\beta_{NI_{SR},HO}$. |
| **Metrics of performance of the Goldberg cutoffs (simulation)** | **Description** |
| $Bias_{NI,HO} = \left(\dfrac{1}{n}\sum_{i=1}^{n}\hat{\beta}_{NI,HO,i}\right) - \beta_1$ | Bias in the estimation of NI-HO association. $\beta_{NI,HO,i}$ is the point estimate from the $i$th simulation |
| $MSE_{NI,HO} = \dfrac{1}{n}\sum_{i=1}^{n}\left(\hat{\beta}_{NI,HO,i} - \dfrac{1}{n}\sum_{i=1}^{n}\hat{\beta}_{NI,HO,i}\right)^2 + \left(\left(\dfrac{1}{n}\sum_{i=1}^{n}\hat{\beta}_{NI,HO,i}\right) - \beta_1\right)^2$ | Mean Squared Error in the estimation of NI-HO association. |
| $P_{NI,HO} = \dfrac{\sum_{i=1}^{n} I\left[\left(\hat{\beta}_{NI,HO,i,low} < \beta_1\right) \cap \left(\beta_1 < \hat{\beta}_{NI,HO,i,high}\right)\right]}{n}$ | Coverage probability in the estimation of NI-HO association. $\beta_{NI,HO,i,low}$ and $\beta_{NI,HO,i,high}$ are lower and upper 95% confidence intervals. |

**Table 2.** Baseline characteristics of the analyzed data[*].

| Variable | IDATA | P-value |
|---|---|---|
| Total Number | 303 | |
| Age (years) | 63.0±5.9 | |
| Male | 124 (40.9%) | |
| Race | | |
| Non-Hispanic White | 283 (93.4%) | |
| African American | 19 (6.3%) | |
| Asian | 1 (0.3%) | |
| Weight (kg) | 79.4±17.1 | |
| Height (cm) | 168.8±9.0 | |
| Waist circumference (cm) | 92.0±14.2 | |
| BMI (kg/m2) | 27.7±4.7 | |
| Fat-free mass (kg) | 48.9±10.9 | |
| Daily EI estimated from ASA24 ($EI_{SR}$; kcal/day) | 2048.0±783.4 | |
| Daily EI estimated from DLW ($EI_{BIO}$; kcal/day) | 2400.3±492.8 | |
| Reporting bias in EI ($EI_{SR}$ - $EI_{BIO}$) | −352.3±811.0 | <0.001 |
| Daily SI estimated from ASA24 ($SI_{SR}$; mg/day) | 3457.9±1440.8 | |
| Daily SI estimated from urine ($SI_{BIO}$; mg/day) | 4015.0±1995.7 | |
| Reporting bias in SI ($SI_{SR}$ - $SI_{BIO}$) | −557.0±2075.2 | <0.001 |
| Daily PoI estimated from ASA24 ($PoI_{SR}$; mg/day) | 2931.4±1136.9 | |
| Daily PoI estimated from urine ($PoI_{BIO}$; mg/day) | 3210.9±1253.9 | |
| Reporting bias in PoI ($PoI_{SR}$ - $PoI_{BIO}$) | −279.5±1404.8 | <0.001 |
| Daily PrI estimated from ASA24 ($PrI_{SR}$; mg/day) | 83.0±38.8 | |
| Daily PrI estimated from urine ($PrI_{BIO}$; mg/day) | 94.5±38.3 | |
| Reporting bias in PrI ($PrI_{SR}$ - $PrI_{BIO}$) | −11.5±41.6 | <0.001 |

*Values are mean ± SD or n (%).

Evaluating the performance of the Goldberg cutoffs solely depending on a single empirical dataset is limited because the dataset is a single realization from an unobserved data-generation process. Further, if the dataset is too small, conclusions may be limited because of issues related to power to detect bias in associations or the power to detect reductions in bias. Therefore, in addition to the empirical data analyses above, we also generated data through simulation and analyzed those generated data that preserve the characteristics of the empirical data. The simulation further enables us to assess the impact of sample size on the performance of the Goldberg cutoffs.

## Results
### Statistical data analysis on IDATA
The list of variables, equations, and metrics is available in *Table 1*. Abbreviations are summarized in *Supplementary file 1*.

**Table 3.** Summary of nutrition intakes of the accepted and the rejected cases by the Goldberg cutoffs*.

|  |  | Accepted | Rejected | P-value[†] |
|---|---|---|---|---|
| Number |  | 183 | 120 |  |
| Bias | Energy intake (kcal/d) | −26±33 | −850±89 | <0.001 |
|  | Sodium intake (mg/d) | −201±142 | −1100±200 | <0.001 |
|  | Potassium intake (mg/d) | 3±98 | −710±129 | <0.001 |
|  | Protein intake (g/d) | −2±3 | −26±4 | <0.001 |
| Self-reported | Energy intake (kcal/d) | 2320±37 | 1633±86 | <0.001 |
|  | Sodium intake (mg/d) | 3757±93 | 3002±145 | <0.001 |
|  | Potassium intake (mg/d) | 3240±71 | 2461±112 | <0.001 |
|  | Protein intake (g/d) | 91±3 | 71±4 | <0.001 |
| Biomarker | Energy intake (kcal/d) | 2346±34 | 2482±49 | 0.023 |
|  | Sodium intake (mg/d) | 3958±141 | 4102±195 | <0.001 |
|  | Potassium intake (mg/d) | 3237±88 | 3171±124 | 0.666 |
|  | Protein intake (g/d) | 93±3 | 100±4 | 0.367 |

*Values are mean ± SD or n.
[†]Mean difference between the accepted cases and the rejected cases was tested.

## The impact of the Goldberg cutoffs on the reporting bias in nutrition intake

*Table 2* summarizes the data included in the analyses (n=303). More than 90% of the participants are non-Hispanic whites, and their mean age was 63. The mean BMI was 27.7 kg/m$^2$, and 30% had obesity (BMI ≥30). There was significant mean underreporting in which the biomarker nutrition intakes ($NI_{BIO}$) was greater than the self-reported nutrition intakes ($NI_{SR}$) in all four types of nutrition intake (*Table 2*).

By the Goldberg cutoffs, 120 among 303 participants were excluded (40%) (*Table 3*). There was significant underreporting in the rejected cases, whereas reporting bias ($NI_{SR} − NI_{BIO}$) was not significant in the accepted cases in all four NI. Further, the difference in the biases between the accepted and the rejected cases was significant. The mean $NI_{SR}$ in the accepted cases was significantly larger than that in the rejected cases in all four NI. $NI_{BIO}$ was significantly larger in the rejected cases in sodium intake (SI), whereas it was comparable between those groups in energy intake (EI), potassium intake (PoI), and protein intake (PrI). To further understand the dependency of the bias on $NI_{BIO}$ and the impact of the Goldberg cutoffs on the bias, the individual data were plotted (*Figure 1*). The bias was significantly and negatively correlated with $NI_{BIO}$ (*Figure 1* **top panels**). The means of accepted cases of $NI_{SR}$ correspond to the mean of $NI_{BIO}$ better than the mean of $NI_{SR}$ in the rejected cases (*Figure 1* **bottom panels**).

These findings suggest that the Goldberg cutoffs reduce (and may eliminate) the bias in mean $NI_{SR}$ by identifying predominantly underreporting individuals. The difference in self-reported nutrition intake between accepted and rejected cases is predominantly from differences in reporting and less so from differences in actual biomarker-measured intake. However, statistically significant differences in biomarker-measured intake remained between accepted and rejected cases in SI.

## Associations between nutrition intake and health outcomes

The associations between $EI_{BIO}$, $SI_{BIO}$, and $PrI_{BIO}$ and WC were statistically significant (*Figure 2—figure supplements 1 and 2* and *Supplementary file 2*). $PoI_{BIO}$ was significantly associated with VO2. Assuming that the biomarker-based health outcome associations ($\beta_{NI_{BIO},HO}$) and self-report-based health outcome associations ($\beta_{NI_{SR},HO}$) are different, the associations of health outcomes with self-report after applying the Goldberg rule ($\beta_{NI_G,HO}$) are expected to be between raw self-report and biomarker-based exposures. In other words, the Goldberg cutoffs are supposed to reduce the bias:

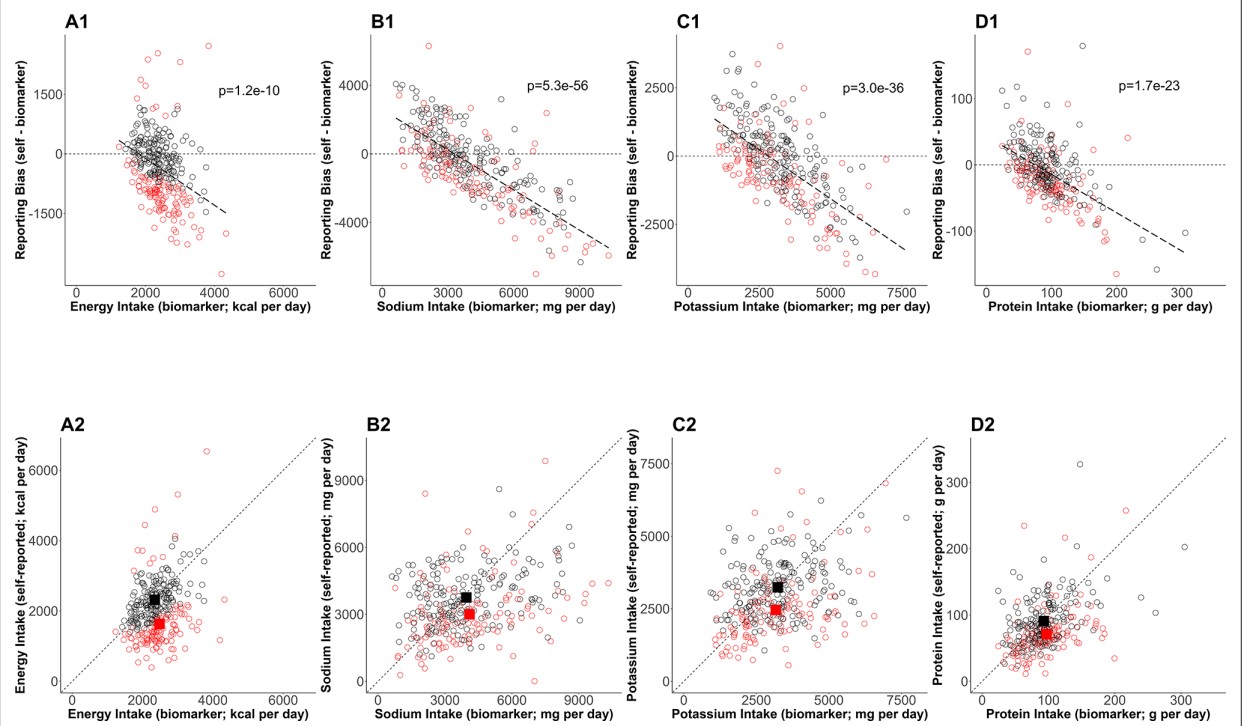

**Figure 1.** Bias in self-reported nutrition intakes. (**Upper panels**) The error in mean of self-reported nutrition intake (**A**: energy intake, **B**: sodium intake, **C**: potassium intake, **D**: protein intake) and relevant nutrition intake measured by biomarkers are plotted. The regression lines are plotted with dashed lines and the corresponding p-values are stated. The dotted horizontal lines at zero indicate there is no error in nutrition intake. The red circles and black circles are the rejected cases and the accepted cases, respectively. (**Bottom panels**) Self-reported nutrition intake and biomarker-based nutrition intake measured are plotted. Closed red and black squares are the mean of nutrition intake in the rejected cases and the accepted cases ($NI_G$), respectively.

($\beta_{NI_{BIO},HO} < \beta_{NI_G,HO} < \beta_{NI_{SR},HO}$ or $\beta_{NI_{BIO},HO} > \beta_{NI_G,HO} > \beta_{NI_{SR},HO}$). We first tested whether bias exists for each nutrition-health outcome pair, and, if so, whether the bias is overestimation or underestimation, using percent bias ($b$). The percent bias ($b$) was significantly below 0% (underestimation) in only limited cases (EI and BW, and EI and WC) and we could not conclude whether the bias exists or not for the rest of the cases. Significant overestimation ($b > 0$) was not observed in any cases (the left panel in *Figure 2* and *Supplementary file 2*). In the two cases of underestimation from self-report (EI and BW, and EI and WC), there were also significant associations between $NI_{BIO}$ and the outcomes. In all cases, the nutrition-health outcome pairs with significant associations between $NI_{BIO}$ and health outcomes (HO) had smaller confidence intervals for the percent bias than those pairs without significant associations between $NI_{BIO}$ and HO (EI and BW, EI and WC, SI and BW, SI and WC, PoI and VO2, PrI and BW, and PrI and WC, shown in ***italic bold*** in *Figure 2*). Second, we tested whether the bias remains after applying the Goldberg cutoffs, and, if so, whether it is overestimation or underestimation, using the percent remaining bias ($r$). The bias was reduced ($|r| < 100$) but remained ($r \neq 0$) for the association between EI and BW, and between EI and WC. We could not again conclude whether the bias exists or not after applying the Goldberg cutoffs for the rest of the cases.

From the above data analyses, we could not conclude that the Goldberg cutoffs do or do not remove the bias in most cases, because of the wide confidence intervals, which could be partially due to the sample size of the data and weak associations between some combinations of NI and HO.

## Simulation

To strengthen the arguments on the performance of the Goldberg cutoffs, we also generated and analyzed data which preserve the characteristics of the IDATA. The simulation was designed to overcome the sample size limitations of IDATA, but keep the structure and nature of the IDATA.

Performance metrics (bias, MSE, and coverage probability) for all the combinations between NI and HO are shown in *Figure 3*. For each performance metric, the mean across the 1000 simulation

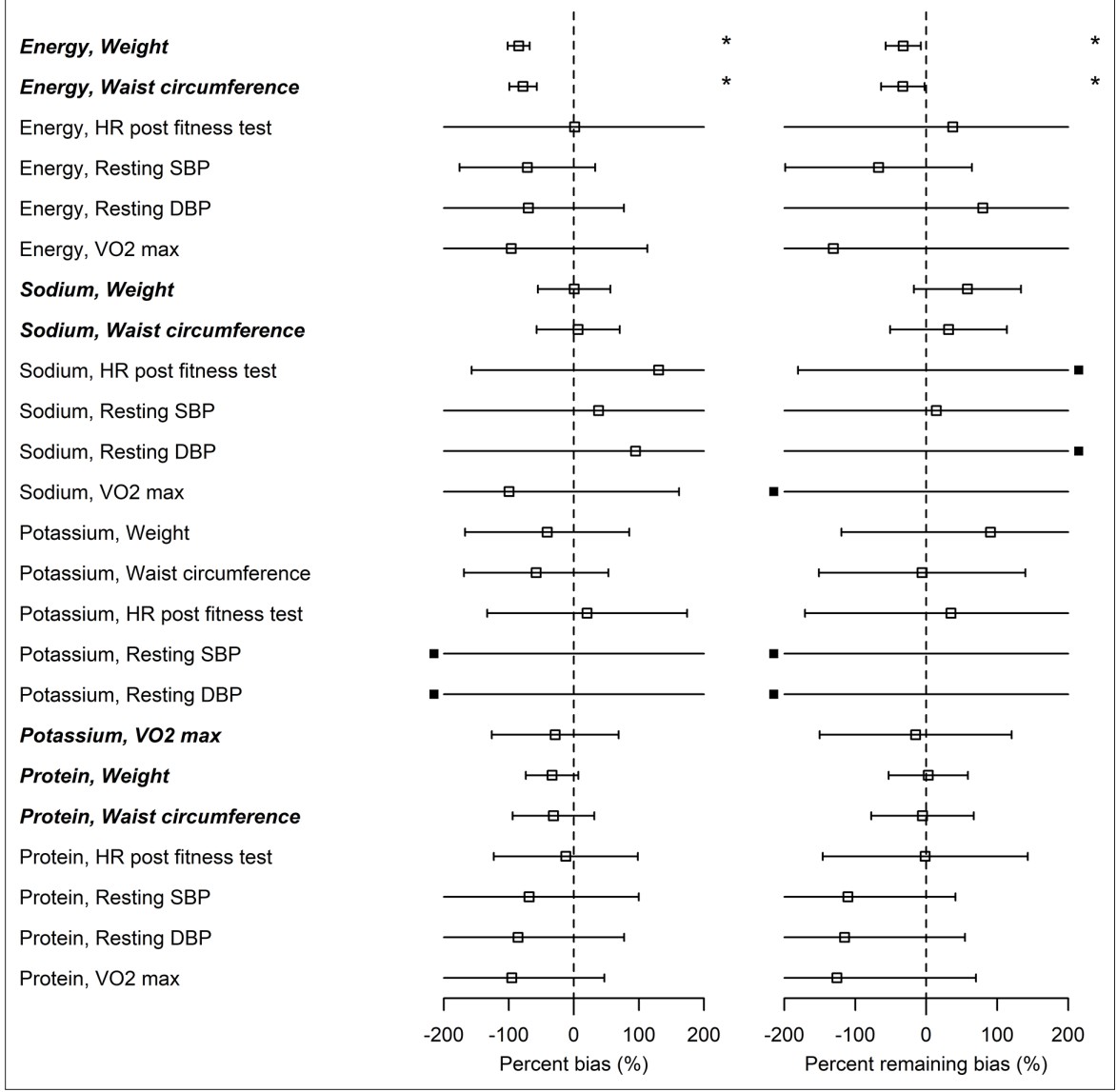

**Figure 2.** Bias in associations between self-reported nutrition intakes and health outcomes. *Italic bold* font denotes significant associations between nutrition intake measured by biomarkers and the outcome (see $\beta_{BIO}$ column in **Supplementary file 2**). Open squares correspond to the maximum likelihood estimators and the bars are 95% CIs. Closed squares are plotted at the left end or right end of the panel when the point estimate is beyond the x-axis limits. Using three types of regression coefficients ($\beta_{SR}$: self-reported data, $\beta_{BIO}$: biomarker data, $\beta_G$: Goldberg accepted data), two metrics of bias were defined: Percent bias of the linear regression coefficient, $b_\beta = (\beta_{SR} - \beta_{BIO})/\beta_{BIO} * 100\ (\%)$ (**Left** panel; *: Significant bias was observed); percent remaining bias of the linear regression coefficient, $d_\beta = (\beta_G - \beta_{BIO})/\beta_{BIO} * 100\ (\%)$ (**Right** panel; #: Significant bias reduction was observed [i.e., bias reduction 95% CI is within –100–100]; *: Significant remaining bias was observed).

The online version of this article includes the following source data and figure supplement(s) for figure 2:

**Source data 1.** Bias in associations between self-reported nutrition intakes and health outcomes.

**Figure supplement 1.** Linear regressions of health outcomes on nutrition intakes.

**Figure supplement 2.** Linear regressions of health outcomes on nutrition intakes.

replicates is shown. We confirmed the generated self-reported data were consistent with the IDATA (**Figure 4—figure supplement 1**). The bias for all biomarker simulations was close to 0 (red squares in the bias column of **Figure 3**). For all four NI, the associations with HO were negative and those with the other outcomes were positive (**Figure 2—figure supplements 1 and 2**). For the cases with positive correlation, the positive bias in $\beta_{NI_{SR},HO}$ was observed, and for the cases with negative correlation, the negative bias in $\beta_{NI_{SR},HO}$ was observed, suggesting that the self-reported data cause attenuation

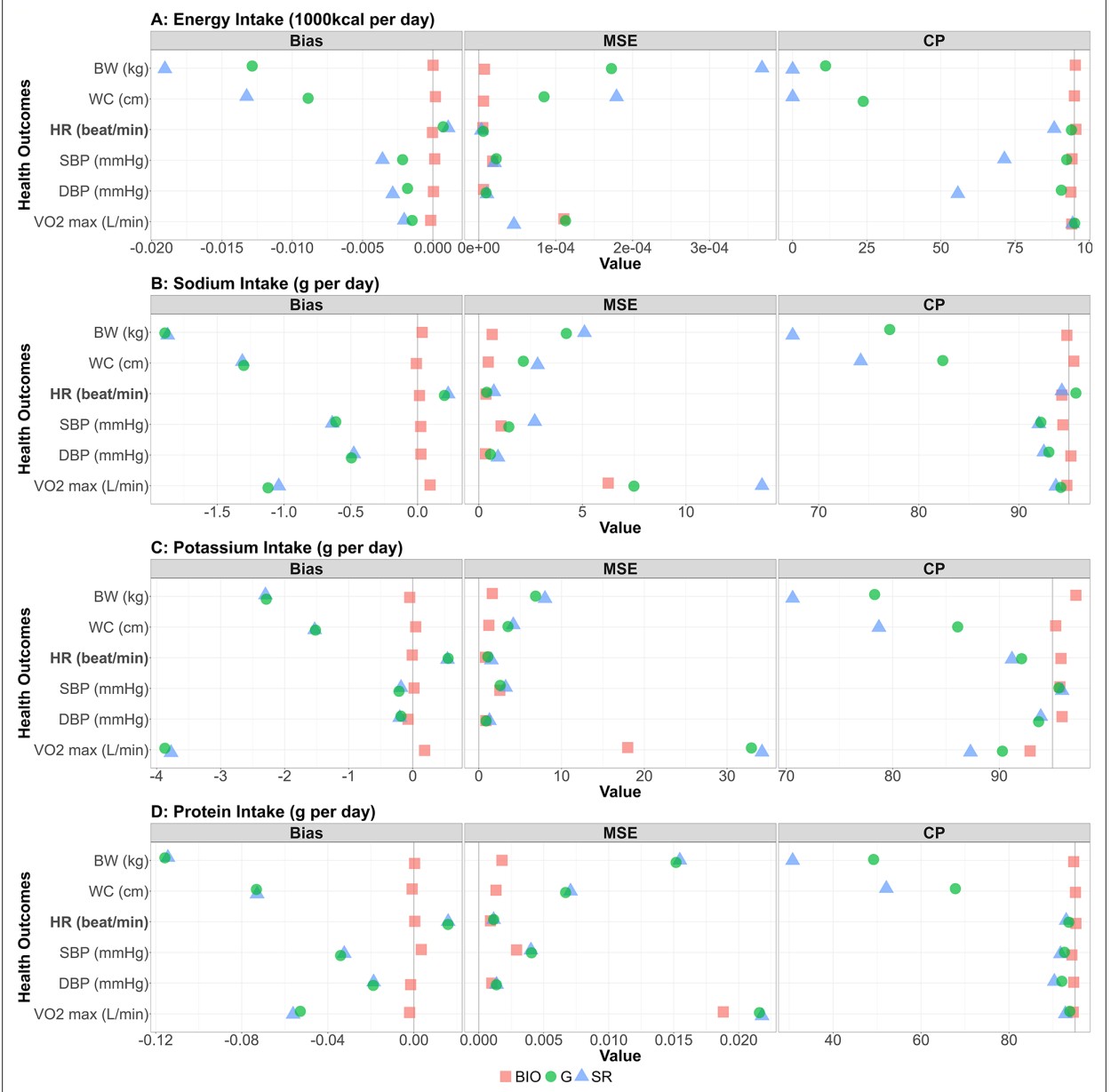

**Figure 3.** Bias, MSE, and coverage probability in the simulation study. The bias, mean squared error (MSE), and the coverage probability (CP) in regression coefficients between four nutrition intakes (**A**: energy intake, **B**: sodium intake, **C**: potassium intake, **D**: protein intake) and six health outcomes (body weight, waist circumference, HR [heart rate] post fitness test, resting SBP [systolic blood pressure], resting DBP [diastolic blood pressure], and VO$_2$ max) for 1000 replicates. Bold font denotes negative associations between nutrition intake measured by biomarkers and the outcome. Red square, blue triangle, and green circle represent biomarker-based nutrition intake, self-reported nutrition intake, and Goldberg accepted nutrition intake, respectively. (**Left panel**) The bias between estimated and true regression coefficients are plotted. The grey vertical line at zero indicates there is no bias between the true regression coefficient and the mean of the 1000 replicates. For each combination, if the green dot is closer to 0 than the blue triangle, then that indicates Goldberg cutoff rule reduced the bias. (**Middle panel**) The MSE between estimated and true regression coefficients are plotted. The grey vertical line at zero indicates there is no bias between the true regression coefficient and the mean of the 1000 replicates. For each combination, if the green dot is closer to 0 than the blue triangle, then that indicates Goldberg cutoff rule reduced the MSE. (**Right panel**) The coverage probability for simulation studies. The grey vertical line indicates coverage probability consistent with 95% confidence intervals. For each combination, if the green dot is closer to 95 than the blue triangle then that indicates the Goldberg cutoff rule improved the coverage probability.

The online version of this article includes the following source data and figure supplement(s) for figure 3:

**Source data 1.** Bias, MSE, and coverage probability in the simulation study.

**Figure supplement 1.** Sensitivities of the sample size relative to the base value.

**Figure supplement 2.** Sensitivities of parameters relative to the base values.

bias (i.e. $|\beta_{NI_{SR},HO}| < |\beta_{NI_{BIO},HO}|$) as observed in many studies (*Kipnis et al., 2003*). After application of the Goldberg cutoffs, the bias was reduced in 14 among 24 combinations of NI and HO, which can be seen in *Figure 3* where the green circles are closer to 0 than the blue triangles. Specifically, $|\beta_{NI_G,HO}| < |\beta_{NI_{SR},HO}|$. Notably, the bias was reduced in all six outcomes and reduction was large when EI is used as a predictor. When SI, PoI, and PrI are used as predictors, the bias was not much changed by applying the Goldberg cutoffs (green circles and blue triangles are close to each other in the bias column of *Figure 3*). In most cases, the MSE was also smaller for NI$_{BIO}$ compared with NI$_{SR}$, and the MSE for NI$_{SR}$ became smaller after applying the Goldberg cutoffs (green circles are often closer to 0 than the blue triangles in the MSE column of *Figure 3*). The coverage probabilities (CP) were close to 95% when NI$_{BIO}$ was used as a predictor, as was expected, whereas it was mostly lower than 95% when NI$_{SR}$ was used. The coverage probability increased by applying Goldberg cutoffs (green circles are often closer to 95 than the blue triangles in the CP column of *Figure 3*); however, it is still lower than 95% in most cases. We repeated the simulation for other values ($n$=50, 100, 200, and 300) of sample size and all cases showed similar results (*Figure 3—figure supplement 1*). We ran sensitivity analyses varying $\eta$, an error term for the regression between NI and HO, and $\sigma$, the standard deviation of the reporting bias (*Figure 3—figure supplement 2*). Biases were not affected by varying $\eta$ for the range of values we considered, suggesting that error in health outcomes (or variance in health outcomes which cannot be explained by variance in biomarker-based nutrition intake) does not influence NI-HO association estimations where NI is used as an independent variable. Bias was negative when self-reported data ($NI_{SR}$ or $NI_G$) were used under the default value of σ, and the magnitude of bias increased as σ increased (i.e., the association was attenuated). However, when the error is extremely small (e.g. σ=0.25 in *Figure 3—figure supplement 2*), the bias becomes positive, which is because the mean error function ($\mu$) was negative at low NI and positive at high NI, thus the association was intensified. Whether or not the bias becomes positive or negative is determined by the balance between the mean error function and random error.

## Discussion

Dietary self-reporting is a basic component of nutrition epidemiology, particularly for studies of nutrition-health relationships. However, reporting bias in nutrition intake is perhaps one of the greatest impediments to understanding the true effect of nutrition on disease. Failure to account for this bias in self-reports can affect the analysis and interpretation of studies designed to assess the influences of nutrition on health. Correlation or regression coefficients estimating associations between self-reported nutrition intakes and health outcomes can be subject to substantial error (*Freedman et al., 2011*; *Gibson et al., 2017*). To address such reporting bias issues, there has been increased interest in the use of statistical models in conjunction with biomarker data (*Freedman et al., 2010*; *Brown et al., 2021*).

In this study, reporting bias in self-reported nutrition intake was demonstrated using both empirical data analysis and a simulation study. We first examined whether the bias in mean nutrition intake and the bias in the associations between nutrition intake and health outcomes are reduced by applying the Goldberg cutoffs to an empirical dataset, IDATA. We confirmed the bias in mean self-reported nutrition intake was eliminated by applying Goldberg cutoffs to exclude extreme reporters (mostly under-reporters). This bias can be mostly explained by differences in reporting between the accepted and rejected cases rather than a difference in actual nutrition intake between the accepted and rejected cases. We further tested whether the associations between health outcomes and nutrition intake are biased when self-reported data are used, and whether the bias is reduced by applying the Goldberg cutoffs. The significant bias for the associations between EI and BW and the association between EI and WC were reduced but remained after applying Goldberg cutoffs. We did not observe significant bias in associations for other nutrition-outcome pairs, and therefore cannot make conclusions regarding the utility of applying the Goldberg cutoffs to reduce bias for these associations in the empirical data.

To overcome the limitations of the data (i.e. sample size and weak associations among many nutrition-outcome pairs), we generated and analyzed data preserving the characteristics of the IDATA for the assessment of the Goldberg cutoffs. We confirmed the existence of bias in estimated associations when the self-reported data were used, and these biases were reduced, but not eliminated,

by applying the Goldberg cutoffs in general. We also confirmed large MSE and low coverage probability when self-reported data were used; MSE and low coverage probability were improved by using Goldberg cutoffs, but were not recovered to the levels of biomarker data. Overall, the empirical data analyses and the simulation study suggest that the Goldberg cutoffs improve the estimates in nutrition-health outcome associations; however, the biases were not completely removed.

This study has several strengths. First, although energy intake has traditionally been studied more with respect to Goldberg cutoffs, herein we also investigated three other nutrition intakes: sodium, potassium, and protein. In our previous study, we used only energy intake as a predictor (*Ejima et al., 2019*). However, given that the Goldberg cutoffs are believed to reduce the bias in nutrition intake broadly, we expanded the approach to these other nutrition intakes, which can be measured by biomarkers (*Livingstone and Black, 2003*). Second, we conducted the simulation in addition to empirical data analyses. There are two advantages of the simulation design. (1) The data generation process was based on the IDATA dataset. Thus, we were able to simulate datasets reflective of those that would be seen in the real world. Both the sample size of the generated dataset and the variables that we considered are reflective of what is often observed in nutrition studies. (2) Simulation can consider different models for the data-generating process. Although our objective was not to disentangle the underlying data generation mechanisms, we were able to generate data that preserve the characteristics of the IDATA.

The limitations of this study are worth noting. In the analysis of IDATA, we used only bivariate associations for nutrition-health outcomes. It is possible that including various covariates or different models may influence the existence or lack of bias in nutrition-health associations. In the data-generation process for the simulations, assumptions were made on parameter settings. The reporting bias was set to be normally distributed. Modeling the bias using other distributions (such as a heavier tailed distribution that may be more consistent with a zero-bounded intake) may improve model fit, and thus generated more realistic data. We simulated the bias with constant variance, consistent with the apparent stability of variance across the intake quantiles in *Figure 4—figure supplement 2* and *Supplementary file 3*; however, modeling the variance as a function of intake quantile may alter the data-generating process. We used the self-reported nutrition intake reported by ASA24, as it reflects recent dietary intake. The bias we observed in this study may be different for other self-report methods. Finally, the simulation approach, although based on the IDATA, was nonetheless generated from a limited sample size. We are not aware of thresholds for sufficient sample size for empirical-data-based simulation, and we have had success with smaller plasmode based sampling in the past (*Ejima et al., 2020*; *Alfaras et al., 2021*); however, it is possible a larger, more representative, or more robust empirical data set may influence both the empirical and simulated results. Further as the data was from a single population in which white and relatively older people are predominant, generalizability of the study finding should be carefully considered.

In conclusion, our study demonstrated that the use of Goldberg cutoffs can effectively reduce or eliminate bias in mean nutrition intake measurements in the IDATA dataset. However, our analysis also revealed that these cutoffs have limited efficacy in addressing bias in the associations between nutrition intake and health outcomes, as both empirical and simulated results showed. Investigators considering excluding extreme reporters in epidemiological studies should consider whether such approaches are likely to be helpful in answering their specific research questions, as our study has shown that such cutoffs are more useful in estimating mean nutrition intakes than in studies of associations between nutrition intake and health outcomes.

## Materials and methods

We conducted a set of empirical analyses to assess the reporting bias due to self-report and examine whether the Goldberg cutoffs reduce reporting bias using the data from The Interactive Diet and Activity Tracking in the American Association of Retired Persons (IDATA) study (*National Cancer Institute, 2018a*). The reporting bias in the mean of nutrition intakes and the associations (regression coefficients) between nutrition intakes and health outcomes were assessed. We also conducted a simulation study to investigate the performance of Goldberg cutoffs using the results from the empirical analyses as ground truth. Note that in data analyses, we assumed the estimates from biomarkers are the true values, since the true values (i.e. population parameters) are unobservable.

Among various nutrition intakes (NI), we considered sodium intake (SI), potassium intake (PoI), and protein intake (PrI) in addition to energy intake (EI). These NI were selected because they can be measured by urine samples as well as self-reports. We denote self-reported NI by $NI_{SR}$: $EI_{SR}$, $SI_{SR}$, $PoI_{SR}$, and $PrI_{SR}$. We denote biomarker-based NI by $NI_{BIO}$: $EI_{BIO}$, $SI_{BIO}$, $PoI_{BIO}$, and $PrI_{BIO}$. The $NI_{SR}$ data remaining after applying the Goldberg cutoffs (hereafter, we call those 'accepted cases') are denoted by $NI_G$: $EI_G$, $SI_G$, $PoI_G$, and $PrI_G$. Health outcomes (HO) that may be associated with NI were selected from the data dictionaries; body weight (BW), waist circumference (WC), heart rate after fitness test (HR), resting systolic blood pressure (SBP), resting diastolic blood pressure (DBP), and $VO_2$ max (VO2). The estimated regression coefficients are denoted by the subscript of the explanatory variables and the health outcomes (e.g., $\beta_{EI_{SR},BW}$ represents the association between self-reported EI and body weight).

## Goldberg cutoffs

The Goldberg cutoffs are an approach to identify those who report low or high energy intake that, if sustained across long periods of time, would be implausible to sustain life (for underreporting) or present weight (for overreporting), by comparing the reported EI and the energy expenditure (EE) predicted from body composition and physical activity level (*Goldberg et al., 1991*). First, EE is predicted as a product of physical activity level (PAL) and basal metabolic rate (BMR): EE =PAL*BMR. PAL is assumed to be 1.75 (*Dugas et al., 2011*). BMR is predicted using fat-free mass (*Cunningham, 1991*): BMR = 370 + 21.6 * FFM. Second, the ratio of $EI_{SR}$ and predicted EE is computed and examined whether it is in a reasonable range: $e^{-2*\frac{S}{100}} < EI_{SR} : EE < e^{2*\frac{S}{100}}$ , where $S$ is defined as: $\sqrt{\frac{CV_{wEI}^2}{d} + CV_{wB}^2 + CV_{tP}^2}$ . $CV_{wEI}$ , $CV_{wB}$ , and $CV_{tP}$ are the within-subject variation in $EI_{SR}$, the within-subject variation in BMR, and the total variation in PAL, and $d$ is the number of days of diet assessment. We set $CV_{wEI} = 23 \ (\%)$ , $CV_{wB} = 8.5 \ (\%)$ , $CV_{tP} = 15 \ (\%)$ , and $d = 7$ following the previous study (*Black, 2000*). Finally, if individuals are not in the range, they were considered under- or overreporters and were excluded from further analyses.

## Data

### Overview of IDATA

The IDATA study was designed 'to evaluate and compare the measurement error structure of diet and physical activity assessment tools against reference biomarkers'. The study participants 'were recruited from a list of AARP [American Association of Retired Persons] members aged 50–74 years residing in and around Pittsburgh, Pennsylvania' and were screened for eligibility by phone interview or clinic visit. The eligibility criteria were speaking English, having internet access, having no major medical issues (such as diabetes, renal/heart failure, and any conditions affecting fluid balance), having no mobility issues, and having BMI between 18.5 and 40.0 (kg/m²). The participants 'visited the study center three times and also completed assigned activities at home over a 12-month period' (*National Cancer Institute, 2018a*) from early 2012 through late 2013 (*National Cancer Institute, 2018a*; *Matthews et al., 2018*). Self-reported dietary/physical activity information, biomarker data to estimate dietary intake, and the data from physical objective monitors to estimate physical activity were collected. In addition, demographic information (age, sex, race/ethnicity) and anthropometric measurements (weight, height, and waist circumference) were obtained at a clinical visit or phone screening.

### Self-reported nutrition intake: $NI_{SR}$

For self-report dietary assessment ($EI_{SR}$, $SI_{SR}$, $PoI_{SR}$, and $PrI_{SR}$), four different approaches were used in IDATA: Automated Self-Administered 24 Hour Dietary Assessment Tool (ASA24) (*National Cancer Institute, 2018b*), Diet History Questionnaire (DHQ-II) (*Diet History Questionnaire, 2022*), 7-day food checklist, and 4-day food record. These SR approaches were conducted at different times during the IDATA study. We used the data of ASA24 in our analyses because they were collected closest in time to the DLW measurements. ASA24 is an online self-administered recall system asking about a 24-hr dietary recall for the previous day, from midnight to midnight, using a dynamic user interface.

## Biomarker-based nutrition intake: NI$_{BIO}$

Three approaches were used for biomarker assessment of NI: DLW, urine, and saliva (not used herein). EI$_{BIO}$ was computed from DLW data, and SI$_{BIO}$, PoI$_{BIO}$, and PrI$_{BIO}$ were calculated from 24-hr urine samples. EI$_{BIO}$ was estimated as a sum of daily total energy expenditure estimated from DLW and daily change in energy stores during the DLW period. Total energy expenditure was calculated using the approach proposed by *Schoeller et al., 1986*, where the respiratory quotient was assumed as 0.86. Daily change in energy stores during the DLW period (about 2 weeks) was computed from the average daily weight change during the period assuming the energy density of body weight was 2380 kcal/kg (*Bhutani et al., 2017*). The urinary values obtained from 24-hr urine samples were converted into nutrition intakes assuming that 81%, 80%, and 86% of consumed nitrogen, potassium, and sodium are excreted in the urine, respectively. Dietary protein was calculated assuming that 16% of protein is nitrogen. A more comprehensive explanation of measuring EI$_{BIO}$, SI$_{BIO}$, PoI$_{BIO}$, and PrI$_{BIO}$ is reported elsewhere (*Park et al., 2018*).

## Health outcomes

From health outcomes (HO) available in IDATA, we selected objective variables that were potentially associated with nutrition intake. BW (kg) and WC (cm) were obtained at clinical visits; HR (beat/min after fitness test), SBP (mmHg), DBP (mmHg), and VO$_2$ max (L/min) were obtained by the modified Canadian Aerobic Fitness Test (mCAFT).

## Summary of the data

In IDATA, NI$_{SR}$, NI$_{BIO}$, and HO were repeatedly measured at different time points. For fair comparison among them, we selected the data measured in the same month; thus, the data of Month 0 and Month 11 were used from Groups 1&3 and Groups 2&4, respectively (*National Cancer Institute, 2018a*). From 1,082 participants, we excluded 779 participants because we did not have information necessary to apply the Goldberg cutoffs (no fat-free mass information [n=725] or NI$_{SR}$ [n=240]) or there was no urine-based nutrition intake data [n=231]. The remaining 303 participants' data were analyzed.

IDATA data were accessible through the Cancer Data Access System (https://biometry.nci.nih.gov/cdas/idata/; downloaded 11/22/2017) after our project proposal was reviewed and approved by the National Cancer Institute (https://biometry.nci.nih.gov/cdas/approved-projects/1702/).

## Statistical analysis of IDATA

Whether the bias in the mean NI$_{SR}$ compared to NI$_{BIO}$ (e.g. calculated as EI$_{SR}$ - EI$_{BIO}$) would be reduced by the Goldberg cutoffs was tested. We first tested whether the mean bias in each NI$_{SR}$ is significantly different from zero. Then, we repeated the same test for the data of the accepted cases for each NI$_G$ (e.g., EI$_G$ - EI$_{BIO}$). We further tested whether the mean NI$_{SR}$ are different between those who were removed (rejected cases; NI$_{SR}$ excluding NI$_G$) and the accepted cases (NI$_G$).

Further, we tested whether the bias in the estimates of associations between NI and HO would be reduced by the Goldberg cutoffs. We computed regression coefficients ($\beta_{NI,HO}$) between HO and each of the three different NI measuring approaches: NI$_{BIO}$, NI$_{SR}$, and NI$_G$. Note that we did not adjust the analyses for any other covariates, because the purpose of this study is to understand if the Goldberg cutoffs could reliably reduce bias rather than to refine associations accounting for covariates. To comprehensively assess the reporting bias and bias reduction in different outcomes with different units, we used standardized metrics for reporting bias and bias reduction, which was originally proposed in our previous study (*Ejima et al., 2019*). In brief, we used the percent bias ($b$) for the assessment of the magnitude of the bias and the percent remaining bias ($r$) for the assessment of the magnitude of the bias remaining after applying the Goldberg cutoffs, which are defined as follows:

$$b = \frac{\beta_{NI_{SR},HO} - \beta_{NI_{BIO},HO}}{\beta_{NI_{BIO},HO}} * 100 \ (\%)$$

$$r = \frac{\beta_{NI_G,HO} - \beta_{NI_{BIO},HO}}{\beta_{NI_{BIO},HO}} * 100 \ (\%)$$

A more in-depth interpretation of the metric is available in our previous study (*Ejima et al., 2019*). Jackknife estimation (leave-one-out) (*Efron and Stein, 1981*) was used to compute 95% CIs of those metrics. All analyses were performed separately for each combination of nutrition intakes and

outcomes. Two-tailed Student's $t$ test was used to test whether mean of a single variable is different from zero. Two-tailed independent Welch's $t$ test was used to test the mean difference of a single variable from two independent groups. The jackknife method was used to test the mean difference of a single variable from two non-independent groups (i.e. whole data vs the data of accepted cases). The type I error rate was fixed at 0.05 (2-tailed).

## Simulation

To strengthen the arguments on the performance of the Goldberg cutoffs, we also generated and analyzed data that preserve the characteristics of the IDATA.

## Data generation

We generated data composed of four variables: $NI_{BIO}$ (biomarker), $NI_{SR}$ (self-reported), HO (health outcomes: BW, WC, HR, SBP, DBP, and VO2), and FFM (fat-free mass). $NI_{BIO}$ and FFM were resampled from the IDATA. HO is computed using a linear model using $NI_{BIO}$ as a predictive variable:

$$HO = a_0 + a_1 NI_{BIO} + \epsilon, \ \epsilon \sim N\left(0, \eta^2\right),$$

where $a_0$, $a_1$, and $\eta^2$ are the model parameters estimated from the IDATA. $NI_{SR}$ is generated by adding an error term (i.e., reporting error), $e$, to $NI_{BIO}$, adapting the approach proposed by **Ward et al., 2019**:

$$NI_{SR} = NI_{BIO} + e,$$

where $e$ is determined by the percentile of $NI_{BIO}$ in the empirical distribution, $p\left(NI_{BIO}\right)$. The reporting error $e$ is assumed to follow a normal distribution parametrized by percentile-specific mean, $\mu\left(p\left(NI_{BIO}\right)\right)$, and a constant standard deviation, $\sigma$: $e \sim N\left(\mu, \sigma^2\right)$. $\mu$ and $\sigma$ were estimated by fitting polynomial models to the IDATA: $\mu\left(p\left(NI_{BIO}\right)\right) = \sum_{k=0}^{K} m_k p\left(NI_{BIO}\right)^k$, where $K$ is the order of the polynomial. The order of the polynomial function was varied from 1 to 5 and we selected the value of $K$ that yields the smallest MSE. We used k-fold cross-validation with 10 folds to calculate the test MSE for each model. This resulted in $K = 1, 3, 3, 5$ for EI, SI, PoI, and PrI, respectively (**Figure 4—figure supplement 2** and **Supplementary file 4**). For example, assuming $EI_{BIO}$ is 3000 kcal, which corresponds to 90.4 percentile, the mean of the reporting error, μ, was –749.6 and the standard deviation of the reporting error, $\sigma$, was 759.1. Therefore his/her $e$ is randomly sampled from the normal distribution: $N\left(-749.6, 759.1\right)$. We further confirmed our assumption of homoskedastic $\sigma$ is reasonable by the Goldfeld Quandt test (**Goldfeld and Quandt, 1965**; **Supplementary file 3**). The Goldfeld Quandt test compares variance in high and low values of a variable; we discarded the middle 20% of the total observations (eliminated the center 61 observations among 303 in total) to define the high and low groups. The simulated data were truncated so that $NI_{SR} > 0$. The variable $NI_G$ was generated by applying the Goldberg cutoffs to $EI_{SR}$ using FFM (thus, $NI_G$ is a subset of $NI_{SR}$). The above process was repeated for each NI. We generated $n = 100$ individuals, and the data generation was repeated 1000 times, leading to 1000 replicates. The data generating process is summarized in **Figure 4**.

## Measurement of the performance of the Goldberg cutoffs

The point estimates of the regression coefficients for an NI and an HO from the $i$th simulation are denoted as $\beta_{NI,HO,i}$ (e.g., $\beta_{EI_{SR},BW,i}$ is the $i$th regression coefficient for self-reported energy intake and body weight). The performance of the Goldberg cutoffs was assessed by three metrics: bias, mean squared error (MSE), and coverage probability:

$$Bias_{NI,HO} = \left(\frac{1}{n}\sum_{i=1}^{n} \hat{\beta}_{NI,HO,i}\right) - \beta_1,$$

$$MSE_{NI,HO} = \frac{1}{n}\sum_{i=1}^{n}\left(\hat{\beta}_{NI,HO,i} - \frac{1}{n}\sum_{i=1}^{n}\hat{\beta}_{NI,HO,i}\right)^2 + \left(\left(\frac{1}{n}\sum_{i=1}^{n}\hat{\beta}_{NI,HO,i}\right) - \beta_1\right)^2,$$

$$P_{NI,HO} = \frac{\sum_{i=1}^{n} I\left[\left(\hat{\beta}_{NI,HO,i,low} < \beta_1\right) \cap \left(\beta_1 < \hat{\beta}_{NI,HO,i,high}\right)\right]}{n},$$

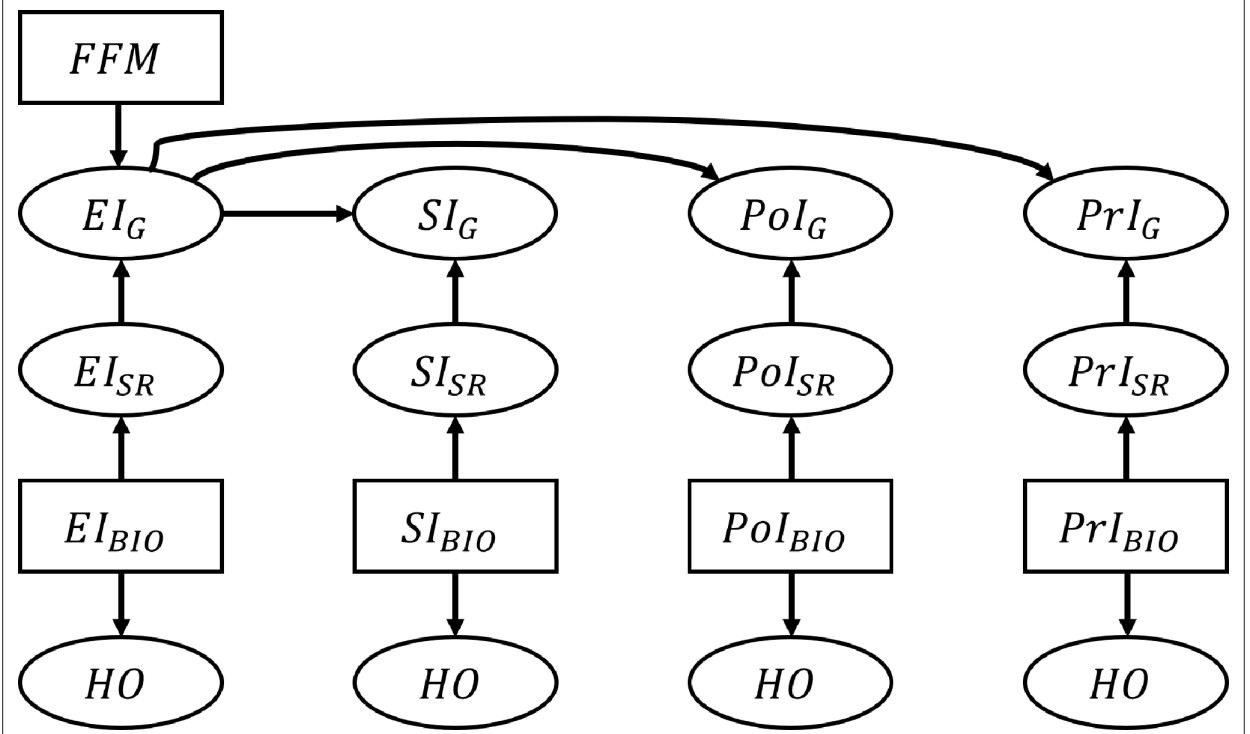

**Figure 4.** Schematic illustration of the data generation process for the nutrition intake and health outcomes. The associations between nutrition intake (EI: Energy Intake, SI: Sodium Intake, PtI: Potassium Intake, PrI: Protein Intake) and health outcomes (HO: body weight, waist circumference, heart rate after fitness test, resting systolic blood pressure, resting diastolic blood pressure, and $VO_2$ max). The subscripts 'SR', 'BIO', and 'G' denote self-reported NI, biomarker-based NI, and self-reported NI after applying the Goldberg cutoffs, respectively. Fat free mass is denoted by FFM and used to calculate the Goldberg cutoff threshold. An arrow from one node, A, to another, B, means 'B is generated by A'. Rectangles represent the variables that are resampled from the empirical distribution, and ellipses are for the variables generated from the models.

The online version of this article includes the following figure supplement(s) for figure 4:

**Figure supplement 1.** Distribution of nutrition intake.

**Figure supplement 2.** Distribution of the reporting error.

where $NI = \{EI_{SR}, EI_{BIO}, EI_G, SI_{SR}, SI_{BIO}, SI_G, PoI_{SR}, PoI_{BIO}, PoI_G, PrI_{SR}, PrI_{BIO}, PrI_G\}$, $HO = \{BW, WC, HR, SBP, DBP, VO2\}$, $I(\cdot)$ is an indicator function, *low* and *high* represent the lower and upper 95% confidence intervals, and $\beta_1$ is the true parameter value, which we define as the point estimate from the IDATA. Note that bias is used differently here than in the empirical analysis because the parameters remain unknown in the empirical analysis whereas we define the parameters in the simulation.

Sensitivity analyses were performed varying the sample size and two other parameters: error term for the regression between health outcome and biomarker-based nutrition intakes ($\eta$), and the standard deviation of the reporting error ($\sigma$). All simulations and analyses were performed using the statistical computing software R (version 4.0.1); code book and analytic code are publicly and freely available without restriction at http://doi.org/10.5281/zenodo.7783842. The variables, data generation models, and metrics of performance of the Goldberg cutoffs are summarized in *Table 1*. STROBE guideline was followed, and the statement is available as supplementary material.

# Additional information

### Competing interests

Roger S Zoh: Reviewing editor, *eLife*. Andrew W Brown: AWB declares: in the past three years (since 2019-09-19), Dr. Brown has received travel expenses from International Food Information Council;

speaking fees from Eastern North American Region of the International Biometric Society, Purchaser Business Group on Health, Purdue University, and University of Arkansas for the Medical Sciences; monetary awards from American Society for Nutrition; consulting fees from LA NORC, Pennington Biomedical Research Center, and Soy Nutrition Institute Global; and grants through his institution from Alliance for Potato Research & Education, American Egg Board, National Cattlemen's Beef Association, NIH/NHLBI, NIH/NIDDK, and NIH/NIGMS. In addition, he has been involved in research for which his institution or colleagues have received grants or contracts from Center for Open Science, Gordon and Betty Moore Foundation, Hass Avocado Board, Indiana CTSI, NIH/NCATS, NIH/NCI, NIH/NHLBI, NIH/NIA, NIH/NIGMS, NIH/NLM, and Sloan Foundation. His wife is employed by Reckitt Benckiser. The other authors declare that no competing interests exist.

### Funding

| Funder | Grant reference number | Author |
| --- | --- | --- |
| National Heart, Lung, and Blood Institute | R25HL124208 | Andrew W Brown |
| Japan Society for the Promotion of Science | KAKENHI grant 18K18146 | Keisuke Ejima |
| National Institute of Diabetes and Digestive and Kidney Diseases | R25DK099080 | Andrew W Brown |
| National Institute of General Medical Sciences | R25GM141507 | Andrew W Brown |
| National Institute of Diabetes and Digestive and Kidney Diseases | 1R01DK132385-01 | Andrew W Brown |
| National Cancer Institute | U01-CA057030-29S1 | Roger S Zoh |
| Meiji Yasuda Life Foundation of Health and Welfare | | Keisuke Ejima |

The funders had no role in study design, data collection and interpretation, or the decision to submit the work for publication.

### Author contributions

Nao Yamamoto, Software, Formal analysis, Investigation, Visualization, Methodology, Writing – original draft; Keisuke Ejima, Conceptualization, Resources, Data curation, Software, Formal analysis, Supervision, Funding acquisition, Validation, Investigation, Visualization, Methodology, Writing – original draft, Project administration; Roger S Zoh, Formal analysis, Supervision, Methodology, Writing – review and editing; Andrew W Brown, Resources, Formal analysis, Supervision, Funding acquisition, Investigation, Methodology, Writing – review and editing

### Author ORCIDs

Nao Yamamoto ⓘ http://orcid.org/0000-0003-3131-2368
Keisuke Ejima ⓘ http://orcid.org/0000-0002-1185-3987
Roger S Zoh ⓘ http://orcid.org/0000-0002-8066-1153

### Decision letter and Author response

Decision letter https://doi.org/10.7554/eLife.83616.sa1
Author response https://doi.org/10.7554/eLife.83616.sa2

## Additional files

### Supplementary files

• Supplementary file 1. List of abbreviation.

• Supplementary file 2. Estimated regression coefficients of the 3 analyses (per Mcal/day for EE and g/day for the others) with 95% CI computed by the jackknife method.

- Supplementary file 3. Statistical tests on the heteroskedasticity of the reporting error.
- Supplementary file 4. MSE of fitted polynomial functions for the reporting error.
- MDAR checklist
- Reporting standard 1. STROBE checklist.

### Data availability

The data can be obtained from NCI after permission, and we are not allowed to redistribute the data. Researchers interested in accessing the original data should submit a project proposal. Specifically, "To gain access to available IDATA data and/or biospecimens, you must submit a project proposal. These are reviewed by NCI. If your project is approved, you will be required to complete a Data Transfer Agreement (and a Material Transfer Agreement, if applicable) before you will be granted access". (https://cdas.cancer.gov/idata/). All simulations and analyses were performed using the statistical computing software R (version 4.0.1). Code book and analytic code will be made publicly and freely available without restriction at http://doi.org/10.5281/zenodo.7783842.

The following dataset was generated:

| Author(s) | Year | Dataset title | Dataset URL | Database and Identifier |
|---|---|---|---|---|
| Ejima A, Yamamoto N | 2023 | Bias in nutrition-health associations is not eliminated by excluding extreme reporters in empirical or simulation studies | http://doi.org/10.5281/zenodo.7783842 | Zenodo, 10.5281/zenodo.7783842 |

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
