## [Editor Report]

This important study estimated the bias in nutrition-health associations and the reduction of bias using Goldberg cut-offs. The evidence supporting the study claims is solid, and the findings will be of interest to epidemiologists and nutrition scientists who are concerned with the effects of measurement error in health-diet research.

---

## [Decision Letter]

**Decision letter after peer review:**

Thank you for submitting your article "Bias in nutrition-health associations is not eliminated by excluding extreme reporters in empirical or simulation studies" for consideration by *eLife*. One peer reviewer has reviewed your article, and I have overseen the evaluation in the dual role of Reviewing Editor and Senior Editor. The reviewer has opted to remain anonymous.

Essential revisions:

As is customary in *eLife*, we discussed the critique with the reviewer. What follows below are the essential and ancillary points and the interaction post-review. Please submit a revised version that addresses these concerns directly. Although we expect that you will address these comments in your response letter, we also need to see the corresponding revision clearly marked in the text of the manuscript. Some of the reviewers' comments may seem to be simple queries or challenges that do not prompt revisions to the text. Please keep in mind, however, that readers may have the same perspective as the reviewers. Therefore, it is essential that you attempt to amend or expand the text to clarify the narrative accordingly.

*Reviewer #1 (Recommendations for the authors):*

Yamamoto et al. analyzed the IDATA data to assess the bias of nutrition intake and the reduction of the bias using the Goldberg cutoffs. They successfully illustrated that the bias reduction is not large in some situations. The data used contain biomarker-based nutrition intake information like the DLW method used as a gold standard in the present study. Although the analyses were precise, the message of the study is quite complicated. A more understandable message would raise the value of the present study.

1. In the abstract, IDATA should be spelled out. Also, please show the name of "IDATA" and it’s spelling out in the introduction section.

2. There are many figures, although the result section of the manuscript does not contain sufficient numerical information; readers are hard to follow the discussion alongside the table and figure. For example, please specify in the text in what situation the bias is especially large or is reduced by the modification method and how large it is.

3. The study concludes that "there is a situation that the Goldberg cutoffs did not solve the problem." However, readers would like to know in what situation the Goldberg cutoffs are not useful and which alternatives are available.

---

## [Author Response]

Reviewer #1 (Recommendations for the authors):Yamamoto et al. analyzed the IDATA data to assess the bias of nutrition intake and the reduction of the bias using the Goldberg cutoffs. They successfully illustrated that the bias reduction is not large in some situations. The data used contain biomarker-based nutrition intake information like the DLW method used as a gold standard in the present study. Although the analyses were precise, the message of the study is quite complicated. A more understandable message would raise the value of the present study.

We appreciate Reviewer #1's positive comments on the precision of our analysis. We also agree that the message of the study may be complicated, and we apologize for any confusion caused by our presentation of the results.

To address this concern, we have revised the manuscript to make the key findings more accessible to a wider audience. Specifically, we have simplified the language used throughout the manuscript and provided more context for our analyses. Additionally, we have emphasized the main takeaways of the study conclusion section of the paper, which we hope will make our message more understandable to readers.

Once again, we appreciate the constructive feedback from Reviewer #1 and believe that our revisions will improve the clarity and accessibility of the manuscript.

1. In the abstract, IDATA should be spelled out. Also, please show the name of "IDATA" and it’s spelling out in the introduction section.

Thank you for this comment. As suggested by the reviewer, we have spelled out IDATA in the abstract and introduced IDATA in the introduction section.

2. There are many figures, although the result section of the manuscript does not contain sufficient numerical information; readers are hard to follow the discussion alongside the table and figure. For example, please specify in the text in what situation the bias is especially large or is reduced by the modification method and how large it is.

We appreciate the reviewer’s comment. We revised the manuscript to provide more clarity in the presentation of results for the empirical study (Page 9, Lines 172-177):

“In the two cases of underestimation from self-report (EI and BW, and EI and WC), there were also significant associations between NI_BIO_ and the outcomes. In all cases, the nutrition-health outcome pairs with significant associations between NI_BIO_ and HO had smaller confidence intervals for the percent bias than those pairs without significant associations between NI_BIO_ and HO (EI and BW, EI and WC, SI and BW, SI and WC, PoI and VO2, PrI and BW, and PrI and WC, shown in *italic bold* in Figure 2).”

We also revised the manuscript of the Results section for the simulation study (Page 10, Lines 198-202):

“After application of the Goldberg cutoffs, the bias was reduced in 14 among 24 combinations of NI and HO, which can be seen in Figure 3 where the green circles are closer to 0 than the blue triangles. Specifically, |βNIG,HO |<|βNISR,HO |. Notably, the bias was reduced in all six outcomes and reduction was large when EI is used as a predictor. When SI, PoI, and PrI are used as predictors, the bias was not much changed by applying the Goldberg cutoffs (green circles and blue triangles are close to each other in Figure 3).”

Similarly, we emphasized the qualitative assessment of the figures by drawing readers to compare the green circles and blue triangles later in that text section and in the figure legend. We hope that this, combined with our original design of keeping the symbols and colors consistent across main and supplementary figures, will help the reader follow the results.

3. The study concludes that "there is a situation that the Goldberg cutoffs did not solve the problem." However, readers would like to know in what situation the Goldberg cutoffs are not useful and which alternatives are available.

We would like to thank the reviewer for this comment, which allowed us to clarify our discussion. Goldberg cutoffs are useful in reducing and in some cases completely eliminating the bias in mean nutrition intake; however, their performance when considering the association between nutrition intake and health outcome is more nuanced (Page 13, Lines 279-282):

“In conclusion, our study demonstrated that the use of Goldberg cutoffs can effectively reduce or eliminate bias in mean nutrition intake measurements in the IDATA dataset. However, our analysis also revealed that these cutoffs have limited efficacy in addressing bias in the associations between nutrition intake and health outcomes, as both empirical and simulated results showed.”

In addition to the conclusion, we also emphasized these in the introduction section (Page 7, Lines 120-122):

“While reducing bias in the mean nutrition intake is useful for determining the true value of population nutrition intake from self-reported data, it is also important to reduce bias in the associations between nutrition intake and health outcomes.”